# Preparation and Characterization of Soap-Free Vinyl Acetate/Butyl Acrylate Copolymer Latex

**DOI:** 10.3390/ma13040865

**Published:** 2020-02-14

**Authors:** Yifu Zhang, Wenkai Bei, Zhiyong Qin

**Affiliations:** School of Resources, Environment and Materials, Guangxi university, Nanning 530000, China; zhyf1026@gxu.edu.cn (Y.Z.); beiwenkai666@163.com (W.B.)

**Keywords:** reactive emulsifier, vinyl acetate, soap-free emulsion

## Abstract

The soap-free emulsion of vinyl acetate (VAc)/butyl acrylate (BA) copolymer was prepared by a semi-continuous and pre-emulsification polymerization method, using ammonium sulfate allyloxy nonylphenoxy poly(ethyleneoxy) (10) ether (DNS-86) as a reactive emulsifier. The effects of DNS-86 on the stability of the emulsion and the properties of the latex film were investigated. The infrared spectrum, thermal stability, glass transition temperature and micromorphology of latex were also studied. The results showed that the emulsion had the best stability and the conversion rate reached a maximum of 98.46% when the DNS-86 amount was 4 wt % of the total amount of monomers. Compared with the PVAc latex synthesized with octylphenol polyoxyethylene ether (10) (OP-10), the latex prepared with DNS-86 has higher thermal stability and ionic stability, whereas the latex film has better water resistance.

## 1. Introduction

Polyvinyl acetate emulsion (PVAc) is a high molecular polymer synthesized by free radical polymerization, which is widely used in industries such as light industry, textile and wood processing [1,2,3]. Generally, the polyvinyl acetate homopolymer has poor water resistance and weather resistance, and the stability of the emulsion is also poor without a protective colloid, so it is often modified with certain methods to improve emulsion performance [4,5,6]. Butyl acrylate (BA) has a long-chain alkyl group which provides a certain steric hindrance. When BA copolymerizes with the VAc monomer, it can protect the ester group of the VAc chain and reduce the hydrolysis rate of the ester group. Thereby, the shortcomings such as water resistance and stability of the polyvinyl acetate emulsion are improved [7,8,9].

The emulsifier has an important influence on the morphology distribution of the latex particles, the polymerization stability of the emulsion, the surface energy and the water resistance of the film. At present, the VAc-BA copolymer emulsion is basically prepared with a conventional emulsifier (ionic or nonionic) in the industry. The emulsifier is mainly composed of a hydrophilic polar group and a lipophilic non-polar group that reduce the interfacial tension of the oil–water system [10,11,12], making the incompatible oil–water system a uniform and stable emulsion system. However, conventional emulsifiers have some disadvantages [13,14,15]. First, since the emulsifier is distributed on the surface of the latex particles by physical adsorption, emulsifier desorption is likely to occur with external action such as low-temperature freezing or mechanical shearing, causing the stable state of the emulsion to be destroyed. In addition, some of the emulsifiers migrate to the surface of the latex film during the drying process of the emulsion, affecting the surface properties and water resistance of the latex film [16,17]. The reactive emulsifier contains a C=C bond which finally can be bonded to the polymer chain by free radical polymerization. It not only achieves the emulsification effect, but also solves the problem of migration of conventional emulsifiers [18].

At present, researchers have used reactive emulsifiers to carry out emulsion synthesis of polyvinyl acetate. Schoonbrood [19] found in VAc/Veova-10/AA, VAc/BA/AA and VAc/MMA/BA systems that some of reactive emulsifier will be buried inside the particles, affecting the stability of the emulsion. Sarkar [20] studied the effect of different emulsifiers on VAc/BA/AA/AM multi-component copolymer emulsions. Experiments showed that reactive emulsifiers help to prepare emulsions with better water resistance and bonding properties. Zhu [21] synthesized a fumaric acid reactive emulsifier, which can produce a VAc/BA/Veova-10/HFBMA quaternary copolymer emulsion with a better stable state and water resistance. Sun [22] has prepared a cationic reactive emulsifier that significantly improves the water resistance and stability of the polyvinyl acetate copolymer emulsion. Zhang [23] obtained the soap-free VAc/BA latex by adding 1 wt % 2-acrylamido-2-methylpropane sulfonic acid (AMPS) as emulsifier with smaller particle size. However, studies on the synthesis of polyvinyl acetate emulsion using allyloxy reactive emulsifiers have not been reported.

In this study, based on the core–shell structure design and self-crosslinking technology, the VAc/BA copolymer emulsion was successfully prepared using ammonium sulfate allyloxy nonylphenoxy poly(ethyleneoxy) (10) ether (DNS-86) as reactive emulsifier. PVA was used as a protective colloid to improve emulsion stability; diacetone acrylamide (DAAM), adipyl hydrazide (ADH) and acrylic acid (AA) were used as a crosslinking monomer to enhance polymer cohesion. The structure of octylphenol polyoxyethylene ether (10) (OP-10) and DNS-86 are shown in Figure 1. The effects of DNS-86 on the properties of the emulsion were investigated.

## 2. Materials and Methods

### 2.1. Materials

Ammonium sulfate allyloxy nonylphenoxy poly(ethyleneoxy) (10) ether (DNS-86), diacetone acrylamide (DAAM) and adipyl hydrazide (ADH) were supplied from Guangzhou Shuangjian Trading Co., Ltd. (Guangdong, China). Poly(vinyl alcohol) (PVA-1788) and octylphenol polyoxyethylene ether (10) (OP-10) were supplied by Beijing Chemical Works (Beijing, China). Vinyl acetate (VAc), butyl acrylate (BA) and acrylic acid (AA) were obtained from the Xilong Chemical Co., Ltd. (Shantou, China). Ammonium persulfate (APS) and phosphotungstic acid (HPWA) were purchased from Tianjin Kemiou Chemical Reagent Co., Ltd. (Tianjing, China). The deionized water was obtained by ion exchange.

### 2.2. Preparation of Emulsion

PVA and deionized water were added to a flask and raised to 90 °C with stirring; after the PVA was completely dissolved, it was cooled to room temperature to obtain PVA solution.

The emulsifier, deionized water, VAc and PVA solution were added to the flask and stirred for 0.5 h at room temperature to obtain a component I. The emulsifier, deionized water and VAc-BA-AA-DAAM mixed monomer was added to the flask and stirred for 0.5 h at room temperature to obtain component II.

Component I and APS solution were added to a four-necked flask equipped with a thermometer, condenser and electric stirrer, and heated up to 70 °C. After the system displayed a blue light, the reaction started. Until the reflux in the condenser tube disappeared, component II and the remaining APS solution were added dropwise in 2–3 h for copolymerization. After the completion of the dropwise addition, the temperature was heated up to 78 °C and maintained for 1 h for full reaction, and then cooled to room temperature. Then, ADH was added with an equimolar ratio with DAAM, and finally, the latex was uniformly stirred and discharged. The component amount is shown in Table 1. The emulsion using OP-10 was also prepared by the above method.

### 2.3. Characterization

#### 2.3.1. Conversion Rate

The final conversion rate was calculated by gravimetric analysis. A certain quantity of latex was cast into a petri dish and dried at 120 °C for 1 h. The conversion rate was calculated by the following formulas.
(1)$$\mathrm{Conversion}\;\mathrm{rate}\;(\mathrm{wt}\;\%)=\frac{m_{3}\times \mathrm{solid}\;\mathrm{content}\left(\%\right)-m_{4}}{m_{5}}\times 100\%=\frac{\left(m_{3}\times \frac{m_{2}-m_{0}}{m_{1}-m_{0}}\times 100\%\right)-m_{4}}{m_{5}}\times 100\%$$
where m_0_ is the weight of the petri dish; m_1_ and m_2_ are, respectively, the weight of latex before and after being dried to a constant weight. m_3_ is the total weight of all the materials put in the flask before polymerization, m_4_ is the weight of materials that cannot volatilize during the drying period and m_5_ is the total weight of monomers.

#### 2.3.2. Polymerization Stability

Coagulate in latex was filtered by filter meshwork (100 M), washed by deionized water and dried to a constant weight at 120 °C.
(2)$$\mathrm{Coagulation}\;\mathrm{ratio}\;(\mathrm{wt}\;\%)=\frac{m_{a}}{m_{b}}\times 100\%$$
where m_a_ is the weight of dried coagulate and m_b_ is the total weight of all the monomers.

#### 2.3.3. Water Absorption Rate

The latex films were naturally dried at room temperature. The weighted latex films were dipped in deionized water at 25 °C for 48 h. Then, the water on the surface of the films was quickly erased by filter papers and the films were weighed again. The water absorption ratio of the films was calculated by the following formula:(3)$$\mathrm{Water}\;\mathrm{absorption}\;\mathrm{ratio}\;(\mathrm{wt}\;\%)=\frac{W_{1}-W_{0}}{W_{0}}\times 100\%$$
where W_0_ and W_1_ are the weight of the films before and after the films absorb water.

#### 2.3.4. Mechanical Stability

The mechanical stability of the latex was tested by the centrifugal machine with the rotational speed of 4000 rpm for 30 min. If flocculation, precipitation and delamination occur, the test is considered to have failed.

#### 2.3.5. Ionic Stability

Calcium ion stability was tested by adding 2 mL CaCl_2_ aqueous solutions (0.5 wt %) into 8 mL latex in a test tube and observed after 48 h. If flocculation, precipitation and delamination occurred, the test was considered to have failed.

#### 2.3.6. Contact Angle

Contact angle was measured by the sessile drop method at room temperature, using a contact angle meter (DSA100E, KYUSS, Germany).

#### 2.3.7. FTIR

A thin latex film dried at room temperature for ATR-IR was directly fixed on a sample frame and measured in the range from 4000 to 400 cm^−1^ by using an FTIR spectrometer (Nicolet iS50, Nico-let Instrument Corporation, Madison, WI, USA). The number of scans was 16; scans resolution was 4.0 cm^−1^.

#### 2.3.8. TEM

The morphology of the latex particles was observed via a high-resolution transmission electron microscope (TECNAI, FEI, Portland, OR, USA). The emulsion was slightly diluted to 1 wt %. The diluted emulsion was dripped onto a special copper mesh and dried, then the membrane on the copper mesh was stained with 2% phosphotungstic acid (HPWA) and dried again.

#### 2.3.9. DSC

The glass transition temperature (Tg) of the latex film was measured using a DSC (404C/3/G, NETZSCH, Bavaria, Germany) in the nitrogen atmosphere. A certain amount of freeze-dried samples (approximately 6–8 mg) was weighted for measuring. The nitrogen flow rate was 40 mL/min, the heating rate was 10 °C/min and the temperature from −40 to 60 °C.

#### 2.3.10. TG

The thermal properties parameter of the samples were tested via thermogravimetric analyzer (TGA) (DTG-60, Shimadzu, Kyoto, Japan). A certain quality of samples (approximately 3–5 mg) was obtained for testing. The heating rate was 10 °C/min from 20 to 600 °C, and the nitrogen flow rate was 50 mL/min.

#### 2.3.11. Latex Particle Size Measurement

After the emulsion was diluted to 2 wt %, the latex particle size and particle size distribution were measured using a laser particle size analyzer (Beckman LS320, Beckman Coulter, UK).

## 3. Results and Discussion

### 3.1. Effect of the DNS-86 Amount on Polymerization

Figure 2 shows the effect of reactive emulsifier DNS-86 amount on conversion rate and emulsion polymerization stability. It can be seen that the conversion rate increases first and then decreases with the increase of the DNS-86 amount. When the amount of DNS-86 increases from 1 wt % (accounting for the mass fraction of the monomer, the same below) to 4 wt %, the monomer conversion rate increases from 83.67% to 98.64%. This is because the emulsifier will form micelles in the system which provide a free radical reaction place for the monomer and catalyst [24,25]. While high conversion cannot be achieved with an insufficient number of active species, with the emulsifier amount increased, more monomers polymerized in the micelles and the final conversion rate of the monomer is increased.

The coagulation ratio was used to judge the polymerization stability of the emulsion [26,27]. With the increase of the DNS-86 amount, the coagulation ratio decreases first and then increases. When the amount of DNS-86 is 1 wt %, the coagulation ratio is 5.95%. This is due to that the low emulsifier concentration in the system was insufficient to the stable latex particles during the reaction, which enhances the collision chance of each particle, resulting in more coagulates. The superior polymerization stability is obtained when the DNS-86 amount is 4 wt %.

### 3.2. Effect of DNS-86 on Mechanical and Ionic Stability of Emulsion

Emulsifiers maintain the stability of the emulsion mainly by electrostatic repulsion (ionic emulsifier) and steric hindrance (nonionic emulsifier) in the system [28,29,30]. It can be seen from Table 2 that the emulsion prepared by the OP-10 emulsion system has poor stability of Ca^2+^, and flocculation occurs shortly after the addition of the CaCl_2_ solution. In the DNS-86 system, the emulsion stability test failed when the DNS-86 amount was 1 wt %. When the DNS-86 amount was too low to cover the latex particles interface, collision between the latex particles in the system was intensified by an external force and the stable state of the emulsion was destroyed. With the emulsifier amount increasing, the emulsifier coverage ratio also increases such that the mechanical and ionic stability of the emulsion is improved. When the emulsifier amount exceeded 3 wt %, superior stability was obtained.

### 3.3. Effect of the DNS-86 Amount on Water Resistance of the Latex Film

Table 3 is the water absorption rate data of the latex film with a different emulsifier. It can be seen that as the DNS-86 amount increases from 1 wt % to 4 wt %, the water absorption rate of the latex film decreases from 20.18% to 6.84%. This is because the size of the latex particles decreases as the emulsifier amount increases, which leads to a denser film with less water penetration. In addition, the reactive emulsifier is bonded to the polymer molecular chain, which avoids the emulsifier migration during the film formation. When the amount of DNS-86 is higher than 4 wt %, the water absorption rate of the latex film increased, Partly because some of the emulsifiers fail to connect to the polymer chain through the C=C bond and migrate to the surface of the latex film after film formation, the hydrophilic properties of the latex film is changed by the hydrophilic sulfonic acid group of DNS-86. However, the water absorption rate of the film made of DNS-86 is always lower than that of OP-10. In addition, the same results were obtained from the contact angle data of the latex film of Figure 3 (when the DNS-86 amount increased from 1 wt % to 4 wt %, the contact angle of the film increased from 61.71° to 77.49°).

### 3.4. Effect of the DNS-86 Amount on Size of Latex Particles

Figure 4 illustrates the effects of the DNS-86 amount on the average particle size of latex. With the DNS-86 amount increasing, the average particle size of latex decreases first and then increases. The average particle size of latex reduces to 90 nm when the DNS-86 amount is 4 wt % and the polydispersity index is 1.01, which signifies that a narrow distribution of the latex particle size. As the DNS-86 amount continues to increase, latex particle size increases extremely. This is because of the self-polymerization of DNS-86, which produces a water-soluble homopolymer, and the lack of emulsifier on the interface of latex particles reduces particle stability. The particles will agglomerate to raise the surface charge density to enhance particle stability, resulting in the increase of particles finally.

### 3.5. FT-IR

From the ATR spectrum of the latex film shown in Figure 5, there is no C=C absorption peak at 1680 cm^−1^, indicating that all monomer and the reactive emulsifier successfully participated in the reaction. The deformation vibration of -CH_3_ appears at a wavenumber of 1432 cm^−1^. The asymmetric stretching and bending vibration of -CH_2_- are at 2936 cm^−1^ and 1373 cm^−1^, respectively. At 1733 cm^−1^, it is a stretching vibration peak of C=O. The stretching and bending vibration absorption peaks of the secondary amide N-H bond at 3332 cm^−1^ and 1538 cm^−1^ and the C-N bond stretching vibration absorption peak at 1239 cm^−1^ indicate that DAAM entered the polymer backbone through the reaction. In addition, the N-H(NH_2_) absorption peak at 3310 cm^−1^ was not observed, indicating that ADH had reacted. The absorption peak appearing at 1669 cm^−1^ is the stretching vibration absorption peak of the C=N bond, indicating that the ketone carbonyl group of DAAM reacts with the hydrazide group of ADH.

### 3.6. DSC

Figure 6 is a DSC curve of the latex film. It can be seen that the latex film has two glass transition temperatures (Tg): Tg_1_ = −5.05 °C and Tg_2_ = 34.8 °C. According to the design of the sample, it is known that the compositions of the core phase polymer and the shell phase polymer are different. According to the Gibbs–Dimarzio formula [31,32,33]: $\frac{1}{\mathrm{Tg}}=\frac{w_{1}}{{\mathrm{Tg}}_{1}}+\frac{w_{2}}{{\mathrm{Tg}}_{2}}+\cdots +\frac{w_{i}}{{\mathrm{Tg}}_{i}}$(w_1_, w_2_, …, w_i_—mass percentage of specific monomers in the polymer composition; Tg_1_, Tg_2_, …, Tg_i_—glass transition temperature of specific monomer homopolymer/°C; Tg—glass transition temperature of copolymer), the theoretical value of the shell phase is −2.15 °C and the core phase is 30 °C. Therefore, it can be inferred that Tg_1_ is the glass transition temperature of the shell phase copolymer, and Tg_2_ is the glass transition temperature of the copolymer of the core phase. This shows that the latex particles have two phase structures. While crosslinking between molecular chains restricts the movement of molecular segments, so the values of Tg_1_ and Tg_2_ are different from the theoretical value. On the other hand, two Tg indicated the phase separation between the core phase polymer and the shell phase polymer. However, the final latex film is visibly transparent, which indicates that the phase separation existing is only a microphase separation.

### 3.7. TGA

Figure 7 shows the TG and DTG (Derivative Thermogravimetric) curves of the latex films synthesized with two emulsifiers. Table 4 shows the specific data of the thermogravimetric test of each sample. It can be seen from Figure 7 and Table 4 that the initial decomposition temperature of the latex film prepared with DNS-86 was 293.5 °C. It is 12.6 °C higher than the initial decomposition temperature (280.9 °C) of the latex film prepared with OP-10. The temperatures of the 50 wt % thermal mass loss of the film obtained by DNS-86 and OP-10 were 340.1 °C and 333.5 °C, respectively, and the residual carbon amounts were 3.2 wt % and 1.5 wt %. The data show that the latex film prepared by the DNS-86 has better thermal stability. This is because DNS-86 is polymerized into the molecular chain. The large-volume flexible side group itself can play a role of steric hindrance and shielding effect, delaying the decomposition reaction of the side chain at high temperature.

In addition, the DTG curve shows that there are two peaks in the decomposition process of the film. This information demonstrates that the thermal decomposition of the material is by two stages. The first stage is mainly the elimination of polymer branches, and the formation of small molecules such as acetic acid. The second stage is mainly the thermal degradation of the C-C backbone by random breaks, which eventually forms a carbonized layer [34,35].

### 3.8. TEM

Figure 8 shows TEM images of the latex particles at different scales. Latex particles colored by HPWA can be clearly seen on the graph. The core phase is completely stained with HPWA. The shell phase exhibits different colors due to the inclusion of other monomers such as BA. The latex particles are spherical or ellipsoidal and randomly dispersed together. The particle diameter was about 90 to 100 nm, complying with the particle size analyzer results. In addition, distinguishable phase separation of the bright and shadow layers can be observed, which confirmed the existence of the core–shell structure latex particles.

## 4. Conclusions

In this study, a soap-free VAc/BA copolymer was successfully prepared by the semi-continuous and pre-emulsification polymerization method. When the DNS-86 amount is 4 wt %, the monomer conversion is 98.64%, the coagulation ratio is 0.34% and the latex has better ionic stability.Compared with the PVAc latex prepared with OP-10, the latex film prepared with DNS-86 has a lower water absorption rate, which means that the addition of reactive emulsifier significantly improves the water resistance of the latex film.FTIR demonstrated that each monomer takes part in copolymerization. A distinguishable phase separation of core phase and shell phases was observed obviously by TEM. DSC showed that the copolymer had two Tg, which is consistent with the design of the core–shell structure. TG indicated that the large-volume flexible side group of DNS-86 raised the thermal stability of the latex.

## Figures and Tables

**Figure 1 materials-13-00865-f001:**
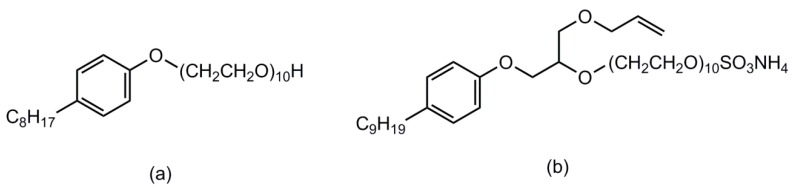
The chemical structures of emulsifiers: (**a**) octylphenol polyoxyethylene ether (10) (OP-10) and (**b**) ammonium sulfate allyloxy nonylphenoxy poly(ethyleneoxy) (10) (DNS-86).

**Figure 2 materials-13-00865-f002:**
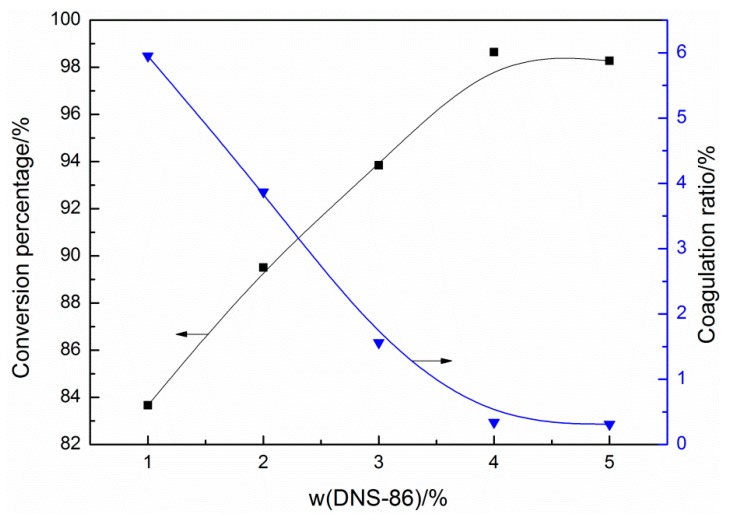
Effect of the DNS-86 amount on conversion rate and coagulation ratio.

**Figure 3 materials-13-00865-f003:**
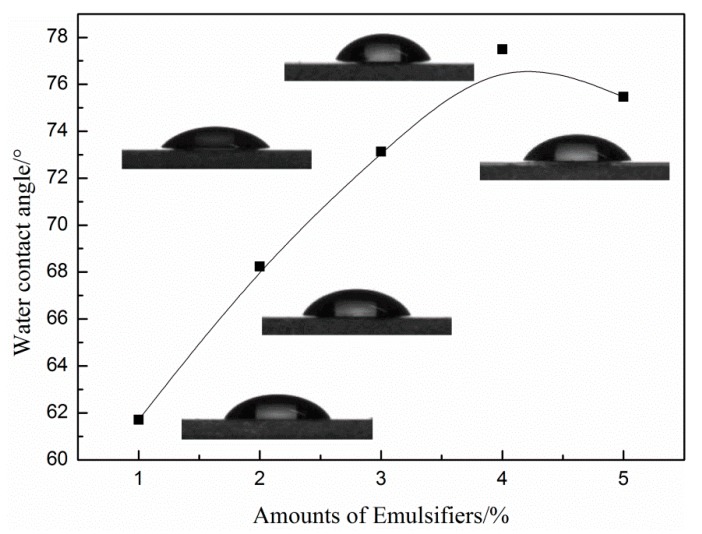
Effect of the DNS-86 amount on water contact angle of the latex film.

**Figure 4 materials-13-00865-f004:**
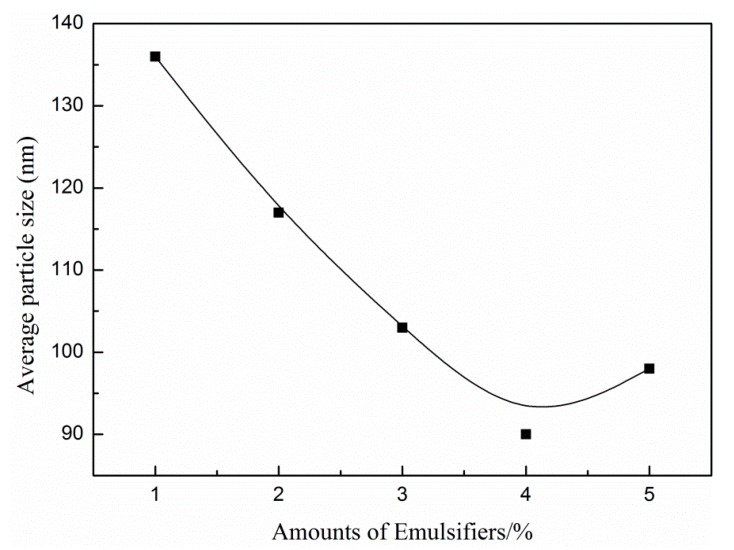
Effects of the DNS-86 amount on particle size.

**Figure 5 materials-13-00865-f005:**
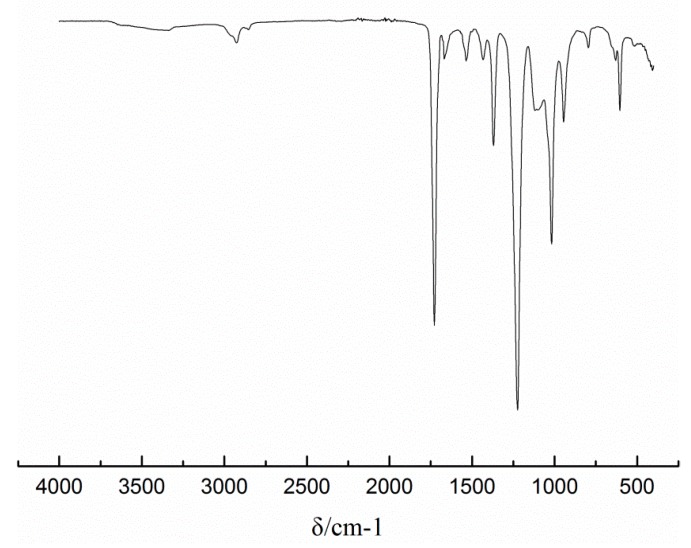
FT-IR spectrum of the latex film with 4% DNS-86.

**Figure 6 materials-13-00865-f006:**
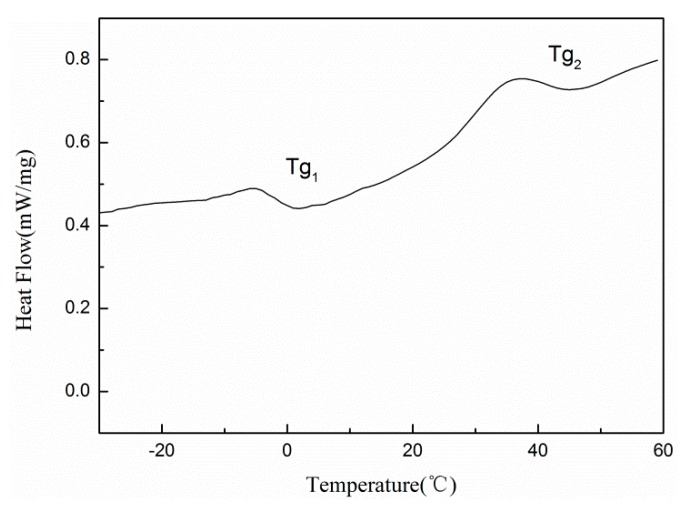
DSC curve of the latex film with 4% DNS-86.

**Figure 7 materials-13-00865-f007:**
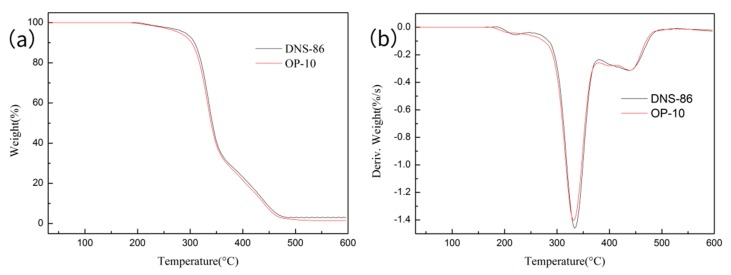
TG (**a**) –DTG (**b**) curve of the latex film with 4% DNS-86 and 5% OP-10.

**Figure 8 materials-13-00865-f008:**
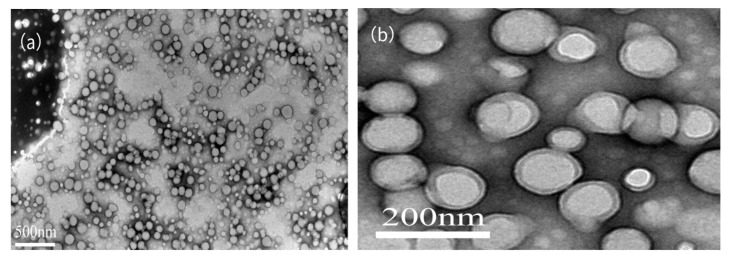
TEM images of emulsion particles with 4% DNS-86: (**a**) 500nm, (**b**) 200nm.

**Table 1 materials-13-00865-t001:** Amounts used in the emulsion polymerization.

	VAc	BA	AA	DNS-86	DAAM	Water	APS	PVA	ADH
Component I/g	30	0	0	0.3–1.2	0	75	0	6	0
Addition/g	0	0	0	0	0	5	0.24	0	0
Component II/g	60	30	1.2	0.9–3.6	2.4	110	0	0	0
Addition/g	0	0	0	0	0	10	0.48	0	2.4
Total/g	90	30	1.2	1.2–4.8	2.4	200	0.72	6	2.4

**Table 2 materials-13-00865-t002:** Effect of the DNS-86 amount on stability of emulsion.

Sample	Amount of DNS-86/wt %					5% OP-10 ^(1)^
	1	2	3	4	5	
Mechanical stability	×	×	√	√	√	√
Ca^2+^ stability	×	×	√	√	√	×

Note: √ means that the test is passed. × means that the test did not pass. ^(1)^—5% is the best amount of experiment.

**Table 3 materials-13-00865-t003:** Effect of the DNS-86 amount on water resistance of the latex film.

Sample	Amount of DNS-86/wt %					5% OP-10
	1	2	3	4	5	
Water absorption rate/%	20.18	16.92	13.47	6.84	11.49	26.81

**Table 4 materials-13-00865-t004:** Thermal degradation temperature of films.

Sample	First Degradation Temperature (°C)			Secondary Degradation Temperature (°C)		
	T_0-1_	T_p-1_	T_f-1_	T_0-2_	T_p-2_	T_f-2_
4%DNS-86	293.5	327.9	340.1	375.3	445.3	510.7
5%OP-10	280.9	325.1	333.5	374.8	444.1	509.5

T_0_: initial degradation temperature; T_p_: maximum degradation rate temperature; T_f_: terminated temperature; T_1_: first degradation; T_2_: second degradation.

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
