# Peer review of "Preparation and Characterization of Soap-Free Vinyl Acetate/Butyl Acrylate Copolymer Latex"

_materials, 2020, doi:10.3390/ma13040865_

Round 1

Reviewer 1 Report

The reviewed article concerns interesting issues of the pre-emulsification polymerization method by using soap-free emulsion of vinyl acetate (VAc)/butyl acrylate (BA) copolymer with new reactive emulsifier 1-allyloxy-3-(4-mercaptophenoxy)-2-propanol polyoxyethylene(10) ether ammonium sulfate (DNS-86).
The authors presented the beneficial effect of the new reactive emulsifier on the stability of the emulsion and on the properties of the received latex films in comparison to the conventional stabilizer.
In my opinion, the reviewed article is characterized by high scientific value and originality. The reviewed manuscript has slight defects that do not reduce the value of the reviewed work.

Below are my comments on the reviewed manuscript.
What was the molecular weight of the PVA used?
The preparation description of component I is not consistent with the information contained in the table.
Please enter the number of scans and scanning resolution in the FTIR description.
Did the drying at 120oC cause material degradation? What was the drying time at this temperature?
How the glass transition temperature was determined?

Author Response

Dear Reviewers:

Thank you for your letter and for the reviewers’ comments concerning our manuscript entitled “Preparation and characterization of soap-free vinyl acetate/butyl acrylate copolymer latex” (ID: materials-701708). Those comments are all valuable and very helpful for revising and improving our paper, as well as the important guiding significance to our researches. We have studied comments carefully and have made correction which we hope meet with approval. Revised portion are marked in red in the paper. The main corrections in the paper and the responds to the reviewer’s comments are as flowing:

Responds to the reviewer’s comments:

Point 1: What was the molecular weight of the PVA used?

Response 1: The molecular weight of PVA is about 84000-89000.

Point 2: The preparation description of component I is not consistent with the information contained in the table.

Response 2:In component I, PVA and APS do not require pre-emulsification, but they are needed for the first step reaction. So it is separate in the preparation description and unified in the table.

Point 3: Did the drying at 120 °C cause material degradation? What was the drying time at this temperature?

Response 3: TG results showed that the material did not begin to decompose until 200 °C, so the material is dried at 120 °C for 1 hours.

Point 4: How the glass transition temperature was determined?

Response 4: The temperature change is controlled by the program, and the relationship between the power difference of the sample and the reference and the temperature is measured, and then the glass transition temperature of the test material is obtained.

We tried our best to improve the manuscript and made some changes in the manuscript.  These changes will not influence the content and framework of the paper. And here we did not list the changes but marked in red in revised paper.

We appreciate for Editors/Reviewers’ warm work earnestly, and hope that the correction will meet with approval.

Thank you very much for your comments and suggestions.

Reviewer 2 Report

The present paper is aimed at investigating the properties of soap free vinyl acetate/butyl acrylate copolymer latex, modified by the addition of a reactive emulsifier.

The subject of the paper is of good interest, and the presented results allow to highlight some of the potential of the proposed approach for improving the stability and the water resistance of the prepared latexes. However, as is the paper does not deserve publication, since some of the presented results are not fully explained, and some more results should be introduced to fully exploit the developed approach. Here is a list of the main issues:

Preparation of the emulsion: it is not clear how the emulsion was prepared, and why some of the components were added. In the title, VAc/BA copolymer are expected to be the subject of this paper. So why AA, DAAm and PVA were also added? What is the specific purpose of each of this components? Also, row 85, what is the initiator solution? And ADH was added in component II, as reported in table I, or at the end of the process? It seems that OP 10 was used as a comparison to the prepared latexes. However, no indication is given of how was the latex with OP 10 prepared, and what is OP 10, apart from its commercial name. Row 99: please provide more details about the derivation of eq. 1) Row 114: how were latex films obtained? Throughout the text, it is not clear when the reaction occurs. Is it expected to occur after addition of component I at 70°C, or after addition of component II at 78? Please explain Row 177: how was gel fraction measured? Row 178: actually no increase of gel fraction is observed at 5%, the difference is quite marginal Mechanical and ionic stability: how was “failing” of the test estimated? Is there a threshold value which allows to define if the test has been passed or not? FTIR analysis: probably it would be better to add the spectra of the single components, in order to verify the existence of the C=C and N-H(NH2) peaks before reaction Row 248: more details about the Gibbs-Dimarzi formula should be given, adding the formula and the numerical values used for calculation Row 261: how was onset calculated? For DNS-86 actually, also a small peak at low temperatures was observed. Did the authors have an explanation for it? The onset for DNS-86 (row 262) should be 293.5 °C and not 239.5°C. I think that the authors should also introduce other significant test, which could improve the quality of the paper. For example, what is the effect of the addition of DNS-86 on the viscosity of the latex, and its evolution during the reaction? I guess that reaction and eventual crosslinking would have a significant effect on the viscosity, and a significant difference would be observed compared to non reactive emulsifiers. Please also check for the English form of the paper (see for example row 179, “this is due to the low emulsifier concentration in the system is insufficient to stable latex particles during reaction”, 224 “and the lacks of emulsifier”, 225, “will agglomeration”

Author Response

Dear Reviewers:
Thank you for your letter and for the reviewers’ comments concerning our manuscript entitled “Preparation and characterization of soap-free vinyl acetate/butyl acrylate copolymer latex” (ID: materials-701708). Those comments are all valuable and very helpful for revising and improving our paper, as well as the important guiding significance to our researches. We have studied comments carefully and have made correction which we hope meet with approval. Revised portion are marked in red in the paper. The main corrections in the paper and the responds to the reviewer’s comments are as flowing:
Responds to the reviewer’s comments:
Point 1: Why AA, DAAM, PVA and ADH were also added?
Response 1: In this system, PVA acts as a protective colloid to increase emulsion stability. AA, DAAM and ADH are used as crosslinking monomers to enhance polymer cohesion. ADH was added at the end of the process.
Point 2: What is the initiator solution?
Response 2: The initiator solution is ammonium persulfate(APS).
Point 3: What is OP-10 and how was the latex with OP-10 prepared?
Response 3: The chemical name of OP-10 is Octylphenol polyoxyethylene ether (10). OP-10 is a common emulsifier. The synthesis method of latex prepared by OP-10 was consist with the latex by DNS-86.
Point 4: How were latex films obtained?
Response 4: All films are naturally dried at room temperature.
Point 5: When the reaction occurs?
Response 5: The reaction is divided into two parts. First step reaction is the self-polymerization of component I at 70 °C. Second step reaction is the multiple copolymerization reaction of component II at 78°C.
Point 6: How was gel fraction measured?
Response 6: Gel fraction is coagulation ratio, used to judge the polymerization stability of the emulsion.
Point 7: How was “failing” of the stability test estimated?
Response 7: When the tested emulsion shows demulsification, the test fails.
Point 8: Add the spectra of the single components?
Response 8: The residual monomer will volatilize at normal temperature, so the existence of C=C cannot be detected by FTIR.
Point 9: How was onset calculated and  how to explain DNS-86 also a small peak at low temperatures was observed?
Response 9: When the mass loss reaches 5%, it is considered to onset calculated. The large-volume flexible side group of DNS-86 just protect part of chain, so the curve of DNS-86 is similar to OP-10.
Point 10: what is the effect of the addition of DNS-86 on the viscosity of the latex, and its evolution during the reaction?
Response 10: This paper is mainly concerned with the study of latex film properties. Research on emulsion viscosity and other issues will be done later.
We tried our best to improve the manuscript and made some changes in the manuscript.  These changes will not influence the content and framework of the paper. And here we did not list the changes but marked in red in revised paper.
We appreciate for Editors/Reviewers’ warm work earnestly, and hope that the correction will meet with approval.
Thank you very much for your comments and suggestions.

Reviewer 3 Report

This paper, focused on the preparation and characterization of soap-free vinyl acetate-butyl acrylate copolymer by pre-emulsification and semi-continuous method, where the effects of DNS-86 on the properties of the emulsion were investigated. The experimental work, based on a combination of several methods, techniques and measurements appear carefully performed and I support publication in this form.

Author Response

Dear Reviewers:

Thanks very much for your kind work and consideration on publication of our paper. On behalf of my co-authors, we would like to express our great appreciation to you.

Reviewer 4 Report

The manuscript materials-701708 demonstrates the development of emulsion stabilizers in the polymerization process and in my opinion can be published in Materials journal. However, I think that the manuscript can be improved for reading. This will increase interest for a wider range of readers. Here are some suggestions:

Abstract should show the essence of the article, in my opinion, and not be a summary of what the authors did. Abstract must be rewritten. It is necessary to indicate the advantages of using DNS-86 in the specific quantitative indicators given in the manuscript. What is the OP-10? Figure 1: chemical structure does not match the name. The chemical structure of OP-10 must also be given. Lines 85-86: What is the "initiator solution", where did they get it? Figure 5: It is necessary to sign the values of the absorption bands directly on the spectrum. Conclusion needs to be improved. It is necessary to indicate the advantages of using DNS-86 in specific quantitative indicators compared to OP-10 and provide an explanation of the identified benefits.

Author Response

Dear Reviewers:

Thank you for your letter and for the reviewers’ comments concerning our manuscript entitled “Preparation and characterization of soap-free vinyl acetate/butyl acrylate copolymer latex” (ID: materials-701708). Those comments are all valuable and very helpful for revising and improving our paper, as well as the important guiding significance to our researches. We have studied comments carefully and have made correction which we hope meet with approval. Revised portion are marked in red in the paper. The main corrections in the paper and the responds to the reviewer’s comments are as flowing:

Responds to the reviewer’s comments:

Point 1: About abstract and conclusion .

Response 1: Abstract and conclusion have been modified for indicating the advantage of using DNS-86.

Point 2:  What is the OP-10?

Response 2:  OP-10 is octylphenol polyoxyethylene (10) ether.

Point 3: DNS-86`s chemical structure does not match the name.

Response 3: The chemical structure of OP-10 and DNS-86 have been shown in new manuscript.

Point 4:  What is the "initiator solution"?

Response 4:  The initiator solution is ammonium persulfate(APS).

We tried our best to improve the manuscript and made some changes in the manuscript.  These changes will not influence the content and framework of the paper. And here we did not list the changes but marked in red in revised paper.

We appreciate for Editors/Reviewers’ warm work earnestly, and hope that the correction will meet with approval.

Thank you very much for your comments and suggestions.

Round 2

Reviewer 2 Report

Unfortunately, only some of the issue raised to the first draft of the paper were addressed. Therefore, I have to suggest rejection of the paper.

Here is a list of the issues not addressed:

Preparation of the emulsion: the specific purpose of the addition of each component was not clarified in the text. The procedure for preparation of the latex is still very confusing No detail about the derivation of eq. 1) is presented No explanation about the times and temperatures of occurrence of the reaction is introduced No detail about the Gibbs-Dimarzi formula has been given No viscosity data is presented.

Author Response

Thank you for your comments.